# Role of Exocrine and Endocrine Insufficiency in the Management of Patients with Chronic Pancreatitis

**DOI:** 10.3390/jcm9062014

**Published:** 2020-06-26

**Authors:** Carmelo Diéguez-Castillo, Cristina Jiménez-Luna, Jose Luis Martín-Ruiz, Joaquina Martínez-Galán, José Prados, Carolina Torres, Amanda Rocío González-Ramírez, Octavio Caba

**Affiliations:** 1Department of Gastroenterology, San Cecilio University Hospital, 18012 Granada, Spain; carmelo89dc@gmail.com (C.D.-C.); jlmartin@ugr.es (J.L.M.-R.); 2Institute of Biopathology and Regenerative Medicine (IBIMER), University of Granada, 18100 Granada, Spain; crisjilu@ugr.es (C.J.-L.); jcprados@ugr.es (J.P.); 3Department of Medical Oncology, Virgen de las Nieves University Hospital, 18014 Granada, Spain; jmgalan22@hotmail.com; 4Department of Biochemistry and Molecular Biology III and Immunology, University of Granada, 18071 Granada, Spain; ctp@ugr.es; 5Instituto de Investigación Biosanitaria (ibs.Granada), 18012 Granada, Spain; agonzalez@fibao.es

**Keywords:** chronic pancreatitis, exocrine pancreatic insufficiency, diabetes mellitus, nutritional status, pancreatic complications

## Abstract

Background: Exocrine pancreatic insufficiency results from the destruction of the pancreatic parenchyma and is diagnosed by using direct or indirect tests, both of which have shortcomings. Chronic pancreatitis is the most frequent cause of this pathology in adults. Methods: Patients meeting radiological or histological diagnostic criteria of chronic pancreatitis are enrolled and the stool elastase test is conducted, considering fecal elastase levels >200 µg/g to represent normal pancreatic function, and levels <200 μg/g to indicate the presence of exocrine pancreatic insufficiency. Additionally, we determine the body mass index of the patients and study their nutritional status and main biochemical and hematological variables, including their glucose and hemoglobin A1c (HbA1c) levels. Results: Exocrine pancreatic insufficiency is detected in 60% of the patients. Among these, 83.3% are severe cases, and 72% of the latter also are diagnosed with endocrine pancreatic insufficiency (diabetes mellitus). During the nutritional status study, HbA1c levels are significantly higher, and magnesium and prealbumin levels are significantly lower in patients with exocrine pancreatic insufficiency than in those without this disease. Conclusions: Exocrine and endocrine pancreatic insufficiency are highly prevalent among patients with chronic pancreatitis and an early diagnosis of these diseases is vital to improve the clinical management of these patients and reduce their risk of mortality.

## 1. Introduction

Exocrine pancreatic insufficiency (EPI) results from the destruction of the pancreatic parenchyma with a sufficiently large loss of acinar cells and/or obstruction of pancreatic ducts that it is not possible to maintain the minimum production levels of digestive enzymes and ductal bicarbonate secretion required to adequately digest food [1,2]. Chronic pancreatitis (CP) is the most frequent cause of EPI in adults [3].

EPI development compromises digestion and the absorption of macro- and micronutrients, and the resulting malnutrition increases the mortality risk [4,5]. Additionally, the presence of EPI significantly increases the risk of complications secondary to CP [6]. Exocrine pancreatic function is diagnosed by using direct or indirect tests, both of which have shortcomings. Direct methods are more sensitive but are only available in a few centers and are laborious, expensive and, being invasive, unpleasant for patients [7]. Indirect methods are less costly, easier to perform, and non-invasive, but their sensitivity for early detection is low [8,9].

The objectives of this study are to analyze the exocrine pancreatic function of patients with CP by using the fecal elastase-1 (FE-1) test during their follow-up and determining their nutritional status and main anthropometric and analytical characteristics.

## 2. Materials and Methods

### 2.1. Study Design and Particpants

We present a cross-sectional study of 50 patients with chronic pancreatitis (CP) under follow-up at San Cecilio University Hospital between February 2015 and June 2016. Inclusion criteria were: age > 18 years; fulfilment of Rosemont diagnostic criteria for CP in patients diagnosed by endoscopic ultrasonography (EUS) [10]; the presence of pancreatic calcifications or dilatation of the main pancreatic duct with parenchymal alterations in those diagnosed by abdominal ultrasound or computed tomography; and a positive histologic result in those diagnosed by pancreatic biopsy. Exclusion criteria were the presence of pancreatic ductal adenocarcinoma (PDAC), refusal to participate in the study, and failure to attend follow-up sessions. PDAC was ruled out by a recent (<6-month) computed tomography, magnetic resonance, or EUS scan.

All patients signed written informed consent to participate in the study, which was conducted in accordance with the Helsinki Declaration and current Spanish legislation, preserving the confidentiality of participants and anonymizing personal data. The protocol was approved by the Ethics Committee of our hospital (approval number 1269-M1-19).

All patients in the pancreatic enzyme replacement therapy (PERT) group were undergoing PERT at the time of laboratory testing and consumed the same amount of enzyme replacement (2 capsules of 25,000 lipase units at breakfast, lunch, and evening meal). Additionally, all patients with a vitamin D deficit were receiving treatment for this condition at the time of testing. Regarding trace elements, all patients underwent a nutritional intervention on healthy life habits, including dietary recommendations to improve the consumption and supply of nutrients.

### 2.2. Assessment of Pancreatic Insufficiency, Nutritional Status, and Pancreatic Complications

Testing was carried out at the time of study enrolment, when an interview was conducted, and a nutritional profile was established. The exocrine pancreatic function of patients was tested using the BioServ Diagnostics Pancreatic Elastase ELISA kit (BioServ Analytics Ltd., Rostock, Germany) to determine fecal elastase-1 (FE-1) levels, following the manufacturer’s instructions. FE-1 ≥200 μg/g was considered to indicate normal pancreatic function, FE-1 = 100–200 μg/g mild-moderate exocrine pancreatic insufficiency (EPI), and FE-1 < 100 μg/g severe EPI. Found at the time of elastase testing, all stools analyzed had a consistency of 1–4 on the Bristol scale. The nutritional status of CP patients with and without EPI was evaluated by determining their body mass index (BMI), their blood concentrations of glucose, glycosylated hemoglobin, total proteins, albumin, prealbumin, somatomedin C, ferritin, transferrin, triglycerides, cholesterol, vitamin D, magnesium, and hemoglobin, and their lymphocyte count.

We studied the presence of pancreatic complications secondary to CP, including compressive (pseudocyst and abscess), stenotic (biliary and duodenal), and vascular (splenoportal axis thrombosis and pseudoaneurysm) complications, the development of type 3c diabetes mellitus (DM)—the diagnostic criterion for diabetes was plasma glucose ≥ 126 mg/dL (7.0 mmol/L) or hemoglobin A1c (HbA1c) ≥ 6.5% (48 mmol/mol)—and the onset of cardiovascular events, defined as major (acute coronary syndrome or stroke) or minor (peripheral arterial disease or deep vein thrombosis). Data on cardiovascular events were gathered by reviewing the clinical records of the Emergency Department and the corresponding Departments of Cardiology, Neurology, and/or Vascular Surgery. Evaluation of chronic abdominal pain was based on the need for analgesics and on the WHO analgesic ladder, referring patients to the Pain Unit of the Anesthesiology Department when the pain was difficult to control. Information on pancreatic abscesses, acute cholangitis, and exacerbation of pancreatitis was collected by reviewing the hospital admissions of patients. The diagnosis of duodenal stenosis was based on the presence of recurrent vomiting and on radiologic and endoscopic findings and was confirmed by biopsy. Pseudocysts and splenoportal axis thrombosis were identified in imaging studies performed every 6–12 months. Finally, the diagnosis of biliary stenosis was based on hepatic profile alterations and cholangio-magnetic resonance imaging (MRI) or endoscopic ultrasonography (EUS) findings.

### 2.3. Statistical Analysis

SPSS v.21.0 (IBM Corp., Armonk, NY, USA) was used for the statistical analyses. After a descriptive analysis, proportions were compared between the groups using the chi-square test or, when conditions were not met, Fisher’s exact test. We applied the non-parametric Mann–Whitney U test to compare mean values of quantitative variables between two groups, and the Kruskal–Wallis test to compare mean values among more than two groups. *p* < 0.05 was considered statistically significant.

## 3. Results

### 3.1. Comparison of Patients with Versus without EPI

Table 1 exhibits the characteristics of the 50 patients with CP included in this study, divided between those with EPI and those without this complication.

CP was defined by characteristic morphological findings, identified with imaging tests in patients with compatible clinical profiles. Most patients (72%) were diagnosed using EUS, based on the Rosemont criteria, while 28% were diagnosed by computed tomography, 6% by abdominal ultrasound, and 4% by pathology report.

Considering the 50 patients with CP, 30 (60%) had FE-1 < 200 μg/g and were, therefore, diagnosed with EPI; in 83.3% of these cases, the EPI was severe (FE-1 < 100 μg/g).

We analyzed the possible association between the presence of EPI and the time interval since CP diagnosis, selecting 12 years as the cutoff point. The time since diagnosis was >12 years in 35% of the patients with EPI and in 5% of those without this complication, although the difference was not statistically significant (*p* = 0.123). No significant association was found between a history of CP for >12 years and being a smoker or alcohol drinker (*p* = 0.174 and *p* = 0.237, respectively).

Body mass index (BMI) values showed that the majority (66.7%) of patients with EPI had normal weight, whereas the majority (60%) of those without EPI were overweight, although the difference did not reach statistical significance (*p* = 0.062). Nutritional status was not compared between patients with mild–moderate and severe EPI due to the wide disparity in sample sizes (25 versus 5).

Shown in Table 2, the patients with and without EPI significantly differed in blood levels of glycosylated hemoglobin (7.16 versus 5.9%, *p* = 0.007), prealbumin (23.64 versus 27.94 mg/dL, *p* = 0.005), and magnesium (1.87 versus 1.98 mg/dL *p* = 0.03); the mean values of these parameters were within the normal range in both groups, except for glycosylated hemoglobin levels, which were above the upper limit of normality in the patients with EPI. Treatment with a proton pump inhibitor was received by 34.4% of the patients with hypomagnesemia. Glucose levels were higher and cholesterol levels were lower in patients with versus without EPI, although statistical significance was not reached (*p* = 0.06 for both variables). Only 12 patients required surgery (resection of pancreas or digestive tract surgery), and they showed no statistically significant difference with the EPI group. No patient in the study was diagnosed with hemochromatosis or celiac disease.

Table 3 exhibits other complications of CP observed in the study population. The most frequent was the presence of pseudocysts (50%), followed by chronic abdominal pain (46%), biliary stenosis (26%), and pancreatitis exacerbation (20%).

### 3.2. Comparison of Patients with EPI on PERT VS. Not on PERT

Results were compared between the patients with EPI who were on pancreatic enzyme replacement therapy (PERT) and those who were not. The only statistically significant between-group difference was the higher frequency of diabetes mellitus (DM) in those on PERT (Table 4).

Comparisons of nutritional status between the patients with EPI on PERT and those not on PERT revealed that glucose and HbA1C values were significantly higher and prealbumin and vitamin D were significantly lower in the group on PERT (Table 5)

No statistically significant differences in CP complications were found between the patients with EPI who were on PERT and those who were not, as shown in Table 6.

### 3.3. Comparison of Patients with Mild/Moderate vs. Severe EPI

Patients were divided between those with mild/moderate EPI and those with severe EPI. The frequency of DM was significantly higher in the severe group, which also had a significantly higher proportion of patients on PERT. No statistically significant differences were observed in the other variables analyzed (Table 7).

Compared to the patients with mild/moderate EPI, glucose and HbA1C values were significantly higher and FE-1 values were significantly lower in those with severe disease. No other statistically significant between-group differences were observed (Table 8).

Shown in Table 9, no statistically significant difference was found between patients with mild/moderate and severe EPI, except for a significantly higher frequency of pseudocysts in those with severe EPI.

## 4. Discussion

The first cause of EPI in adults is the presence of CP, followed by cystic fibrosis and pancreatic surgery [3]. During this study of patients with CP under follow-up at our hospital, we used FE-1 as diagnostic test for EPI because it is non-invasive, easily applied, economic, and offers a high positive predictive value in advanced stages. However, the main limitations of this test are its low sensitivity at earlier stages of EPI and its false positive rate, which may be influenced by the soft consistency of feces or the presence of small intestine disease. The likelihood of false positives was reduced in our study by excluding fecal samples that were not solid [1,5].

An EPI prevalence of 60% was found, similar to the prevalence of 64.1% reported in the “PANCR-EVOL” multi-center study of Spanish patients with CP [11]. During the present investigation, 83.3% of the patients with EPI had FE-1 levels <100 μg/g, indicating an advanced stage of the disease. Similarly, a study of Chilean patients with CP found a comparable prevalence (66.9%) [12], whereas another study in Spain reported a lower prevalence of 38.8% [13].

During the present study, 18% of the patients diagnosed with EPI were not under treatment for this complication and were immediately started on treatment with PERT. Conversely, and similar to the findings of the PANCR-EVOL study [11], PERT was being received by 35% of the patients without EPI. These data underscore the need for an accurate diagnosis of EPI in patients with CP, given the increased morbidity and mortality risk if adequate treatment is not applied [14].

It has previously been reported that most patients with CP had EPI after a median of 12 years post-diagnosis [2]; hence, we set this value as the cutoff point to analyze the differences between groups with and without EPI after this time. Although there was a higher percentage of patients with a CP history of >12 years in the group with versus without EPI, the difference did not reach statistical significance. However, other studies report the development of EPI after a shorter history of CP [15].

During the present study, only 23.3% of the patients with EPI had never been drinkers and only 36.7% had never been smokers. The consumption of tobacco and alcohol is considered to be a prime cause of CP [16], and the synergetic effects of these substances have been found to produce major stress in the endoplasmic reticulum of pancreatic acinar cells, inducing their death [17]. There was a trend in our sample toward a higher proportion of alcohol drinkers in the group with versus without EPI (*p* = 0.051).

The presence of EPI is associated with a poor absorption of macronutrients and micronutrients, making these patients especially prone to malnutrition. Nutritional markers related to CP include fat-soluble vitamins, vitamin B12, zinc, magnesium, calcium, and folic acid [1], and it has been proposed that their combination may assist in the diagnosis of EPI [2]. We used the most frequent analytical and anthropometric parameters to assess the nutritional status of patients.

During our study, we observed no significant association between BMI and the presence of EPI, although the majority of the patients without EPI were overweight, while the majority of those with EPI were normal weight, and it was previously reported that patients with CP are more frequently normal or overweight than underweight [18]. However, it should be borne in mind that BMI provides a limited evaluation of body composition [19], and some of our patients with “normal” weight showed signs of protein malnutrition, with a major loss of muscle mass (sarcopenia) [20].

Compared to the patients without EPI, those with EPI evidenced significantly higher levels of glycosylated hemoglobin. Significantly lower levels of prealbumin and magnesium were observed in those with EPI, indicating a possible malabsorption of macronutrients (proteins) and micronutrients (trace elements) in these patients. Mean magnesium levels in the group without EPI were 2.001 mg/dL, very similar to those obtained in the study by Lindkvist et al., (2.09 mg/dL) [2].

Vitamin D was selected from among the fat-soluble vitamins because it is routinely analyzed in primary care and hospital centers as a known marker of malnutrition. Additionally, coagulation levels serve as an indirect marker of vitamin K and were normal in all study participants. We found no statistically significant differences between the groups regarding vitamin D, or in albumin, somatomedin C, triglycerides, hemoglobin, ferritin, transferrin, or lymphocytes. However, there was a trend toward lower cholesterol levels and higher glucose levels in the patients with versus without EPI, suggesting that a significant difference might emerge with a larger sample size. Hirano et al., (2014) divided patients with CP between those with and without type 3c DM and observed significantly lower total cholesterol levels in the former group (164 mg/dL versus 183 mg/dL, respectively, *p* = 0.0028) [21]. Seen in our sample, the frequency of DM was higher in the group with EPI, although statistical significance was not reached. As expected in patients with CP, which is associated with massive islet destruction, the most frequent type of DM was 3c [22]. Additionally, the presence of DM of any type in our study was higher in the group with EPI, although the difference was not statistically significant (60% in patients with EPI versus 35% in those without, *p* = 0.083). However, as noted above, glycosylated hemoglobin levels were significantly higher in the group with EPI (*p* = 0.007). These data indicate that exocrine and endocrine pancreatic insufficiency develops in parallel.

The prevalence of DM was higher in our study (50%) than in multi-center studies with a larger sample size, such as that by Schwarzenberg et al., (30%), which also found a significant difference in DM prevalence between children and adults, explained by the time taken for the disease to develop and the role of tobacco and alcohol in its etiology [23]. The higher prevalence in our investigation may be due in part to a greater motivation to participate in the study among patients with CP and functional loss (EPI or DM) than among patients with PC who are asymptomatic and have no signs of malnutrition or DM development, who would be less aware of their disease.

Seen in our study population, the most frequent complications of CP were the presence of pseudocysts (50%), chronic pain (46%), biliary stenosis (26%), and pancreatitis exacerbation (20%), as also observed in the PANCR-EVOL study [11], although the rate of these complications was higher in the present population.

Compared to the patients with EPI who were not on PERT, the frequency of DM was significantly higher in those undergoing this therapy, whose glycemic control was worse (higher glycemia and HbA1C values), and their magnesium and prealbumin levels were significantly lower. These findings can be attributed to a positive association between the development of EPI and that of DM, which is corroborated by the higher frequency of DM in the group of patients with severe EPI. Concerning this, Hardt et al., proposed that DM is not a direct cause of EPI but rather a consequence of the CP that also produced EPI, through the destruction of both exocrine and endocrine pancreatic tissue [24]. These data support the idea that the pancreas functions as a whole, and that exocrine and endocrine disorders inevitably have repercussions on each other [25].

Finally, in comparison to patients with mild/moderate EPI, the frequencies of diabetes mellitus and pseudocyst-related complications were significantly higher in those with severe EPI, whose FE-1 levels were significantly lower, as expected. These findings suggest that the clinical management of this type of patient should include a close follow-up protocol to facilitate early diagnosis and treatment and to avoid the development of associated complications.

## 5. Conclusions

This study contributes evidence on the prevalence of EPI and DM in patients with CP. Analysis of their nutritional status revealed altered nutritional variables in the patients, which may be useful information for their clinical management. We also report on differences between patients with EPI who were undergoing PERT and those who were not, and between patients with mild/moderate EPI and those with severe EPI. A finding of interest was the higher frequency of DM observed in the patients who were undergoing PERT. The main limitation was the relatively small sample size, as in many studies on this issue, although it was sufficient for statistically significant differences to be detected. Additionally, there were major disparities in sample size for the comparisons as a function of PERT and EPI severity. Further research in wider samples and external validation studies are needed to determine the clinical relevance of these results.

## Figures and Tables

**Table 1 jcm-09-02014-t001:** Patient characteristics (*n* = 50).

	Total	Patients with EPI	Patients without EPI	*p*-Value
*n*	50	30 (60%)	20 (40%)	
Mean age	54.82	55.77	53.4	0.751
Gender				0.021 ^a^
Male	41 (82%)	28 (93.3%)	13 (65%)	
Female	9 (18%)	2 (6.7%)	7 (35%)	
Caucasian	50 (100%)	30 (100%)	20 (100%)	
BMI (kg/m^2^)				0.062 ^a^
Overweight (>25)	21 (42%)	9 (30%)	12 (60%)	
Normal weight	28 (56%)	20 (66.7%)	8 (40%)	
Underweight (<18)	1 (2%)	1 (3.3%)	0 (0%)	
Diabetes mellitus	25 (50%)	18 (60%)	7 (35%)	0.083
Type 1–2	12 (48%)	10 (55.6%)	2 (28.6%)	
Type 3c	13 (52%)	8 (44.4%)	5 (71.4%)	
>12 years since diagnosis	8 (16%)	7 (23.3%)	1 (5%)	0.123
Diagnostic test				0.233
EUS	36 (72%)	19 (63.3%)	17 (85%)	
CT	14 (28%)	11 (36.7%)	3 (15%)	
Ultrasound	3 (6%)	3 (10%)	0 (0%)	
Pathology report	2 (4%)	2 (6.7%)	0 (0%)	
CP etiology				0.608
Toxic	33 (66%)	22 (73.3%)	11 (55%)	
Idiopathic	11 (22%)	5 (16.7%)	6 (30%)	
Autoimmune	4 (8%)	2 (6.7%)	2 (10%)	
Pancreas divisum	2 (4%)	1 (3.3%)	1 (5%)	
Alcohol habit				0.051
Drinker (SDU)	33 (66%) (7.1)	23 (76.7%) (6.6)	10 (50%) (7.9)	
Smoking habit				0.311
Smoker (cigarettes/day)	39 (78%) (20.5)	19 (63.3%) (20.1)	14 (70%) (22.1)	
PERT	28 (56%)	21 (70%)	7 (35%)	0.015 ^a^
Surgical treatment	12 (24%)	10 (20%)	2 (4%)	0.163

EPI: exocrine pancreatic insufficiency; BMI: body mass index; EUS: endoscopic ultrasonography; CT: computed tomography; CP: chronic pancreatitis; SDU: standard drink unit; PERT: pancreatic enzyme replacement therapy. ^a^
*p* < 0.05.

**Table 2 jcm-09-02014-t002:** Analytical parameters in the study population, group with EPI, and group without EPI.

	Normal	Total	Patients with EPI	Patients without EPI	*p*-Value
*n*		50	30	20	
Glucose (mg/dL)	75–115	116.98 (±51.6)	127.23	104.65	0.06
HbA1C (%)	3–6	6.59 (±1.6)	7.16	5.9	0.007 ^a^
Total proteins (mg/dL)	6.6–8.3	7.19 (±0.5)	7.16	5.9	0.471
Albumin (g/dL)	3.5–5.2	4.19 (±0.4)	4.21	4.28	0.568
Prealbumin (mg/dL)	20–40	25.29 (±6.4)	23.64	27.94	0.005 ^a^
Somatomedin C (μg/L)	81–225	161.08 (±77.2)	143.12	177.37	0.113
Cholesterol (mg/dL)	140–200	191.57 (±47.5)	181.41	199.45	0.06
Triglycerides (mg/dL)	89–150	138.69 (±73.8)	134.8	162.06	0.846
Vitamin D (ng/dL)	20–100	16.49 (±8.9)	14.41	19.53	0.097
Magnesium (mg/dL)	1.8–2.6	1.93 (±0.21)	1.87	1.98	0.03 ^a^
Hemoglobin (g/dL)	12–17.2	14.57 (±1.85)	14.56	14.96	0.649
Ferritin (ng/mL)	20–250	142 (±48.12)	136.14	171.68	0.915
Transferrin (mg/dL)	10–360	284.28 (±63.6)	284.86	290.84	0.562
Absolute lymphocytes	1100–4500	2170.8 (±720)	2340	2375	0.762
Fecal elastase-1 (μg/g)	>200	180 (±185.5)	42.9	386	0.000 ^a^

^a^*p* < 0.05. Data are reported as mean (standard deviation) with the exception of fecal elastase-1 data, for which mean values alone are given. EPI: exocrine pancreatic insufficiency; HbA1C: hemoglobin A1c.

**Table 3 jcm-09-02014-t003:** Complications of chronic pancreatitis.

	Total	Patients with EPI	Patients without EPI	*p*-Value
*n*	50	30 (60%)	20 (40%)	
CP complications				0.137
Patients without complications	10 (20%)	6 (20%)	4 (20%)	
Patients with complications	40 (80%)	24 (80%)	16 (80%)	
Type of complication				
Cardiovascular events	4 (8%)	2 (6.7%)	2 (10%)	0.528
Chronic abdominal pain	23 (46%)	15 (50%)	8 (40%)	0.487
Pseudocyst	25 (50%)	15 (50%)	10 (50%)	0.507
Abscess	2 (4%)	1 (3.3%)	1 (5%)	0.623
Biliary stenosis	13 (26%)	10 (33%)	3 (15%)	0.249
Duodenal stenosis	1 (2%)	0 (0%)	1 (5%)	0.379
Splenoportal axis thrombosis	3 (6%)	3 (10%)	0 (0%)	0.268
Acute cholangitis	1 (1%)	1 (3.3%)	0 (0%)	0.619
Pancreatitis exacerbation	10 (20%)	4 (13.3%)	6 (30%)	0.163

EPI: exocrine pancreatic insufficiency.

**Table 4 jcm-09-02014-t004:** Patient characteristics in patients with EPI on PERT and those not on PERT.

	Total Patients with EPI	Patients with EPI on PERT	Patients with EPI Not on PERT	*p*-Value
*n*	30 (60%)	21 (70%)	9 (30%)	
Mean age	55.77	54.81	58	0.428
Gender				1.000
Male	28 (93.3%)	19 (90.5%)	9 (100%)	
Female	2 (6.7%)	2 (9.5%)	0 (0%)	
Caucasian	30 (100%)	21 (100%)	9 (100%)	
BMI (kg/m^2^)				0.124
Overweight (>25)	9 (30%)	4 (19%)	5 (55.6%)	
Normal weight	20 (66.7%)	16 (76.2%)	4 (44.4%)	
Underweight (<18)	1 (3.3%)	1 (4.8%)	0 (0%)	
Diabetes mellitus	18 (60%)	15 (71.4%)	3 (33.3%)	0.050 ^a^
Type 1–2	10 (55.6%)	7 (46.7%)	3 (100%)	
Type 3c	8 (44.4%)	8 (53.3%)	0 (0%)	
>12 years since diagnosis	7 (23.3%)	7 (100%)	0 (0%)	0.071
Diagnostic test				0.458
EUS	19 (63.3%)	12 (57.1)	7 (77.8%)	
CT	6 (20%)	4 (19%)	2 (22.2%)	
Ultrasound	3 (10%)	3 (14.3%)	0 (0%)	
Pathology report	2 (6.7%)	2 (9.5%)	0 (0%)	
CP etiology				0.583
Toxic	22 (73.3%)	14 (66.7%)	8 (88.9%)	
Idiopathic	5 (16.7%)	4 (19%)	1 (11.1%)	
Autoimmune	2 (6.7%)	2 (9.5%)	0 (0%)	
Pancreas divisum	1 (3.3%)	1 (4.8%)	0 (0%)	
Alcohol habit				0.918
Drinker (SDU)	23 (77%) (6.6)	15 (71.4%) (6.7)	8 (88.9%) (6.6)	
Smoking habit				0.716
Smoker (cigarette/day)	19 (63%) (20.1)	13 (61.9%) (20.6)	6 (66.7%) (20.0)	
Surgical treatment	10 (20%)	9 (42.9%)	1 (11.1%)	0.204

EPI: exocrine pancreatic insufficiency; PERT: pancreatic enzyme replacement therapy; BMI: body mass index; EUS: endoscopic ultrasonography; CT: computed tomography; SDU: standard drink unit. ^a^
*p* < 0.05.

**Table 5 jcm-09-02014-t005:** Analytical parameters in patients with EPI on PERT and those not on PERT.

	Normal	Total Patients with EPI	Patients with EPI on PERT	Patients with EPI Not on PERT	*p*-Value
*n*		30	21	9	
Glucose (mg/dL)	75–115	127.23 (±58.3)	138.57	100.78	0.037 ^a^
HbA1C (%)	3–6	7.16 (±1.8)	7.61	6.01	0.027 ^a^
Total proteins (mg/dL)	6.6–8.3	7.16 (±0.5)	7.15	7.17	0.928
Albumin (g/dL)	3.5–5.2	4.21 (±0.4)	4.19	4.27	0.776
Prealbumin (mg/dL)	20–40	23.64 (±6.8)	21.15	28.18	0.010 ^a^
Somatomedin C (μg/L)	81–225	143.12 (±65.6)	141.17	147.5	0.802
Cholesterol (mg/dL)	140–200	181.41 (±50.8)	178.5	187.22	0.758
Triglycerides (mg/dL)	89–150	134.8 (±66.7)	129.28	149	0.628
Vitamin D (ng/dL)	20–100	14.41 (±6.1)	12.78	17.83	0.019 ^a^
Magnesium (mg/dL)	1.8–2.6	1.87 (±0.2)	1.86	1.91	0.556
Hemoglobin (g/dL)	12–17.2	14.56 (±2.1)	14.21	15.38	0.213
Ferritin (ng/mL)	20–250	136.14 (±301)	166.61	303.23	0.483
Transferrin (mg/dL)	10–360	284.86 (±57.9)	281.7	291.89	0.555
Absolute lymphocytes	1100–4500	2340 (±711)	2308.1	2413.33	0.700
Fecal elastase-1 (μg/g)	>200	42.9 (±12.5)	32.38	74.11	0.132

^a^*p* < 0.05. Data are reported as mean (standard deviation) with the exception of fecal elastase-1 data, for which mean values alone are given. EPI: exocrine pancreatic insufficiency; PERT: pancreatic enzyme replacement therapy.

**Table 6 jcm-09-02014-t006:** Complications of chronic pancreatitis in patients with EPI on PERT and not on PERT.

	Total Patients with EPI	Patients with EPI on PERT	Patients with EPI Not on PERT	*p*-Value
*n*	30 (60%)	21 (70%)	9 (30%)	
CP complications				
Patients without complications	6 (20%)	4 (19%)	2 (22.2%)	0.509
Patients with complications	24 (80%)	17 (81%)	7 (77.8%)	
Type of complication				
Cardiovascular events	2 (6.7%)	2 (9.5%)	0 (0%)	0.506
Chronic abdominal pain	15 (50%)	13 (61.9%)	2 (22.2%)	0.109
Pseudocyst	15 (50%)	12 (57.1%)	3 (33.3%)	0.405
Abscess	1 (3.3%)	1 (4.8%)	0 (0%)	1.000
Biliary stenosis	10 (33%)	8 (38.1%)	2 (22.2%)	1.000
Duodenal stenosis	0 (0%)	0 (0%)	0 (0%)	
Splenoportal axis thrombosis	3 (10%)	2 (9.5%)	1 (11.1%)	1.000
Acute cholangitis	1 (3.3%)	0 (0%)	1 (11.1%)	1.000
Pancreatitis exacerbation	4 (13.3%)	3 (14.3%)	1 (11.1%)	1.000

EPI: exocrine pancreatic insufficiency; PERT: pancreatic enzyme replacement therapy.

**Table 7 jcm-09-02014-t007:** Patient characteristics of patients with mild/moderate versus severe EPI (*n* = 30).

	Total Patients with EPI	Patients with Mild/Moderate EPI	Patients with Severe EPI	*p*-Value
*n*	30 (60%)	5 (16.7%)	25 (83.3%)	
Mean age	55.77	60.8	54.76	0.373
Gender				1.000
Male	28 (93.3%)	5 (100%)	23 (92%)	
Female	2 (6.7%)	0 (0%)	2 (8%)	
Caucasian	30 (100%)	5 (100%)	25 (100%)	
BMI (kg/m^2^)				0.803
Overweight (>25)	9 (30%)	2 (40%)	7 (28%)	
Normal weight	20 (66.7%)	3 (60%)	17 (68%)	
Underweight (<18)	1 (3.3%)	0 (0%)	1 (4%)	
Diabetes mellitus	18 (60%)	0 (0%)	18 (72%)	0.006 ^a^
Type 1–2	10 (55.6%)		10 (55.6%)	
Type 3c	8 (44.4%)		8 (44.4%)	
>12 years since diagnosis	7 (23.3%)	0 (0%)	7 (28%)	0.304
Diagnostic test				0.324
EUS	19 (63.3%)	5 (100%)	14 (73.7%)	
CT	6 (20%)	0 (0%)	6 (24%)	
Ultrasound	3 (10%)	0 (0%)	3 (12%)	
Pathology report	2 (6.7%)	0 (0%)	2 (8%)	
CP etiology				0.879
Toxic	22 (73.3%)	4 (80%)	18 (72%)	
Idiopathic	5 (16.7%)	1 (20%)	4 (16%)	
Autoimmune	2 (6.7%)	0 (0%)	2 (8%)	
Pancreas divisum	1 (3.3%)	0 (0%)	1 (4%)	
Alcohol habit				0.347
Drinker (SDU)	23 (76%) (6.6)	4 (80%) (8.8)	19 (76%) (6.2)	
Smoking habit				0.372
Smoker (cigarette/day)	19 (63%) (20.1)	3 (60%) (15.0)	16 (64%) (21.0)	
PERT	21 (70%)	2 (40%)	19 (76%)	0.143
Surgical treatment	10 (20%)	3 (60%)	7 (28%)	0.300

EPI: exocrine pancreatic insufficiency; BMI: body mass index; EUS: endoscopic ultrasonography; CT: computed tomography; SDU: standard drink unit; PERT: pancreatic enzyme replacement therapy. ^a^
*p* < 0.05.

**Table 8 jcm-09-02014-t008:** Analytical parameters in patients with mild/moderate versus severe EPI. (*n* = 30).

	Normal	Total Patients with EPI	Patients with Mild/Moderate EPI	Patients with Severe EPI	*p* Value
*n*		30	5	25	
Glucose (mg/dL)	75–115	127.23 (±58.3)	89.6	134.76	0.018 ^a^
HbA1C (%)	3–6	7.16 (±1.8)	5.5	7.58	0.009 ^a^
Total proteins (mg/dL)	6.6–8.3	7.16 (±0.5)	7.48	7.09	0.264
Albumin (g/dL)	3.5–5.2	4.21 (±0.4)	4.26	4.2	0.907
Prealbumin (mg/dL)	20–40	23.64 (±6.8)	26.03	22.98	0.325
Somatomedin C (μg/L)	81–225	143.12 (±65.6)	150.6	141.33	0.845
Cholesterol (mg/dL)	140–200	181.41 (±50.8)	172	183.55	0.827
Triglycerides (mg/dL)	89–150	134.8 (±66.7)	111.2	140.7	0.684
Vitamin D (ng/dL)	20–100	14.41 (±6.1)	17.56	13.72	0.126
Magnesium (mg/dL)	1.8–2.6	1.87 (±0.2)	1.94	1.86	0.388
Hemoglobin (g/dL)	12–17.2	14.56 (±2.1)	14.64	14.54	0.956
Ferritin (ng/mL)	20–250	136.14 (±301)	440.46	161.02	0.487
Transferrin (mg/dL)	10–360	284.86 (±57.9)	270.4	287.88	0.751
Absolute lymphocytes	1100–4500	2340 (±711)	2024	2402.76	0.420
Fecal elastase-1 (μg/g)	>200	42.9 (±12.5)	157.4	20	0.001 ^a^

^a^*p* < 0.05. Data are reported as mean (standard deviation) with the exception of fecal elastase-1 data, for which mean values alone are given. EPI: exocrine pancreatic insufficiency.

**Table 9 jcm-09-02014-t009:** Complications of chronic pancreatitis in patients with mild/moderate and severe EPI. (*n* = 30).

	Total Patients with EPI	Patients with Mild/Moderate EPI	Patients with Severe EPI	*p*-Value
*n*	30 (60%)	5 (16.7%)	25 (83.3%)	
CP complications				0.583
Patients without complications	6 (20%)	1 (20%)	5 (20%)	
Patients with complications	24 (80%)	4 (80%)	20 (80%)	
Type of complication				
Cardiovascular events	2 (6.7%)	0 (0%)	2 (8%)	0.138
Chronic abdominal pain	15 (50%)	3 (60%)	12 (48%)	1.000
Pseudocyst	15 (50%)	2 (40%)	13 (52%)	0.039 ^a^
Abscess	1 (3.3%)	0 (0%)	1 (4%)	1.000
Biliary stenosis	10 (33%)	2 (40%)	8 (32%)	1.000
Duodenal stenosis	0 (0%)	0 (0%)	0 (0%)	
Splenoportal axis thrombosis	3 (10%)	2 (40%)	1 (4%)	0.108
Acute cholangitis	1 (3.3%)	1 (20%)	0 (0%)	1.000
Pancreatitis exacerbation	4 (13.3%)	2 (40%)	2 (8%)	1.000

EPI: exocrine pancreatic insufficiency. ^a^
*p* < 0.05.

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
