# Peer review of "Role of Exocrine and Endocrine Insufficiency in the Management of Patients with Chronic Pancreatitis"

_jcm, 2020, doi:10.3390/jcm9062014_

Round 1
Reviewer 1 Report
The additions add strength to the paper. It adds to a more complete story.
Author Response
Response 1: We are grateful for this positive evaluation.
Reviewer 2 Report
This is a third submission of a manuscript after extensive editing.
The authors present data on exocrine insufficiency and its impact on nutritional status in a cohort of patients with chronic pancreatitis. As mentioned before the subject is of high clinical and scientific interest. The new submission substantially improves on the initial submission, mainly by better defining inclusion/exclusion criteria and presenting the data in a clearer fashion.
The concerns this reviewer brought forward with regards to prior submissions are fully answered by the current submission.
Author Response
Response 1: We are very grateful for the invaluable suggestions of this reviewer throughout the review process.
This manuscript is a resubmission of an earlier submission. The following is a list of the peer review reports and author responses from that submission.
Round 1
Reviewer 1 Report
The authors made changes but the biggest problem (low number of included patients) still remains. This is a repetitive type of study: there are a lot of similar studies in the literature and we did not get any new information.Author Response
We agree that the sample size is relatively low, in common with many other studies on this topic, including some multi-center investigations. Nevertheless, it proved sufficient to allow conclusions to be drawn based on statistically significant results. We have been able to establish the prevalence and severity of exocrine pancreatic insufficiency in patients with chronic pancreatitis. We have also demonstrated the potential of the fecal elastase test as a reliable, rapid, non-invasive, and inexpensive diagnostic technique in advanced stages of the disease. As a novel feature, we studied the nutritional status of these patients and detected alterations in nutritional variables (e.g., magnesium and prealbumin) that may be useful for the clinical management of these patients. Our data on complications associated with CP were similar to previous reports but an even higher rate was detected, attributable to the regular follow-up imaging studies conducted at our center. For this reason, we recommend a close follow-up with imaging techniques in these patients.
We have added the following paragraph on the strengths and limitations of our study in the revised Discussion section (page 11; lines 265-273):
“This study contributes evidence on the prevalence of EPI in patients with CP, showing a high proportion of severe cases. Analysis of their nutritional status revealed altered nutritional variables in the patients, which may be useful information for their clinical management. We also report on differences between patients with EPI who were undergoing PERT and those who were not and between patients with mild/moderate EPI and those with severe EPI. The main limitation was the relatively small sample size, as in many studies on this issue, although it was sufficient for statistically significant differences to be detected. In addition, there were major disparities in sample size for the comparisons as a function of PERT and EPI severity. Further research in wider samples and external validation studies are needed to determine the clinical relevance of these results.”
Reviewer 2 Report
This is a resubmission on a priorly submitted paper. As noted in the last review, the topic of exocrine pancreatic insufficiency in chronic pancreatitis is of both scientific and clinical intrest.
The authors clarified a number of methodological issues, which increases the interpretability of the presented data.
Unfortunatly, even after revision, the authors do not statistically compare patients with exocrine pancreatic insufficiency on pancreatic enzyme replacement therapy (PERT) with those not on PERT. One would suspect that patients with exocrine pancreatic insufficiency had a worse nutritional status if not on PERT, and combining both patients with and without PERT at the time of data collection is most likly masking differences in comparison of patients with and without pancreatic inusfficiency. The reason for this decision remain unclear.
Author Response
We have now performed statistical analysis to compare between patients with exocrine pancreatic insufficiency who were on PERT and those who were not. The results are exhibited in the revised Results (pages 5-7; lines 148-168; Tables 4-6) and are discussed in the revised Discussion (page 11; lines 254-259).
Reviewer 3 Report
- The mean amount of enzyme replacement should be listed between the two groups as adequate dosing may be a confounder
- Why were most patients diagnosed with CP on EUS? That is puzzling as most should have adequate CT/MR imaging prior that are adequate. Is this a selection bias?
- It would be helpful to delineate mild/mod from severe EPI for Table 2 if enough patients are available for analysis. There is a clinical significance if the means are highly driven by the severe EPI pts (which I suspect since FE-1 levels are very low in the EPI group)
- In Table 2: SD should also be listed for all factors including FE-1
- Table 3 needs p-values
Other:
- There are a few spelling errors throughout the manuscript
- PERT needs to be fully written out prior to abbreviation
- To clean up Table 1: please list just EtOH drinker and Smoker status (no need for non-drinker and non-smoker rows as that is implied)
Author Response
Point 1: The mean amount of enzyme replacement should be listed between the two groups as adequate dosing may be a confounder.
Response 1: All patients in both groups received the same amount of enzyme replacement (2 capsules of 25,000 lipase units at breakfast, lunch, and evening meal). This information has now been added in the revised Materials and Methods (page 2; lines 69-70).
Point 2: Why were most patients diagnosed with CP on EUS? That is puzzling as most should have adequate CT/MR imaging prior that are adequate. Is this a selection bias?
Response 2: All patients were diagnosed by CT/MR imaging; however, when the imaging findings were not sufficient to declare a definitive diagnosis, this was confirmed by EUS.
Point 3: It would be helpful to delineate mild/mod from severe EPI for Table 2 if enough patients are available for analysis. There is a clinical significance if the means are highly driven by the severe EPI pts (which I suspect since FE-1 levels are very low in the EPI group).
Response 3: As recommended by the reviewer, we conducted a statistical analysis to compare between patients with mild/moderate EPI and those with severe EPI, exhibiting the findings in the revised Results (page 7-10; lines 169-190; Tables 7-9), and discussing them in the revised Discussion (page 11; lines 260-264).
Point 4: In Table 2: SD should also be listed for all factors including FE-1.
Response 4: The SD for FE-1 is now reported in Table 2.
Point 5: Table 3 needs p-values
Response 5: Table 3 now displays the p-values.
Point 6: There are a few spelling errors throughout the manuscript.
Response 6: The manuscript has been thoroughly re-revised by a professional editor (US English style).
Point 7: PERT needs to be fully written out prior to abbreviation.
Response 7: This has been done (page 2; line 68).
Point 8: To clean up Table 1: please list just EtOH drinker and Smoker status (no need for non-drinker and non-smoker rows as that is implied).
Response 8: Table 1 has been modified accordingly.